# PATH-SPECIFIC CAUSAL FAIR PREDICTION VIA AUXILIARY GRAPH STRUCTURE LEARNING

## ABSTRACT

Algorithm fairness has become a trending topic, and it has a great impact on social welfare. Among different fairness definitions, path-specific causal fairness is a widely adopted one with great potentials, as it distinguishes the fair and unfair effects that the sensitive attributes exert on algorithm predictions. Existing methods based on path-specific causal fairness either require graph structure as the prior knowledge or have high complexity in the calculation of path-specific effect. To tackle these challenges, we propose a novel casual graph based fair prediction framework which integrates graph structure learning into fair prediction to ensure that unfair pathways are excluded in the causal graph. Furthermore, we generalize the proposed framework to the scenarios where sensitive attributes can be non-root nodes and affected by other variables, which is commonly observed in real-world applications but hardly addressed by existing works. We provide theoretical analysis on the generalization bound for the proposed fair prediction method, and conduct a series of experiments on real-world datasets to demonstrate that the proposed framework can provide better prediction performance and algorithm fairness trade-off.

## 1 INTRODUCTION

With the ubiquitous adoption of machine learning algorithms to facilitate decision making, algorithm fairness has attracted increasingly more attentions, in the areas such as recommendation system Ge et al. (2021); Zhu et al. (2018); Burke (2017); Yao & Huang (2017), natural language processing Bolukbasi et al. (2016); Zhao et al. (2017); Gonen & Goldberg (2019); De-Arteaga et al. (2019); Blodgett et al. (2020), computer vision Shankar et al. (2017); Nagpal et al. (2019); Raji et al. (2020); Stock & Cisse (2018), hiring Hoffman et al. (2018), education Brunori et al. (2012), banking Mukerjee et al. (2002), and crime risk assessment Brennan et al. (2009); Dieterich et al. (2016); Zhang & Bareinboim (2018b). Algorithm fairness aims to reduce or even eliminate *unjustified distinctions of individuals based on their sensitive attributes (e.g., gender and race) during the prediction* Zhang & Wu (2017). Unfortunately, machine learning models constructed from the raw data are vulnerable to the unfairness risk due to the historical prejudices in the data. It is crucial for model designers to take algorithm fairness into consideration for long-term social welfare.

In recent years, researchers have developed a variety of causal fairness definitions to help machine learning models make fair predictions Zhang et al. (2017); Huang et al. (2020); Nabi & Shpitser (2018); Kusner et al. (2017); Russell et al. (2017); Wu et al. (2019a); Zhang & Bareinboim (2018a;b); Hu et al. (2020); Xu et al. (2019); Wu et al. (2018); Zhang et al. (2016), and one of them, path-specific causal fairness Chiappa (2019); Nabi & Shpitser (2018); Wu et al. (2019b), is adopted in this paper. Under the definition of path-specific causal fairness, unfairness is viewed as the presence of *unfair causal effect through the disallowed causal pathway* that the sensitive attributes exert on predictions. In other words, a fair prediction satisfies path-specific causal fairness if it eliminates the causal effect that the sensitive attributes assert on the prediction through disallowed causal pathways. Such a definition provides the flexibility of tracing the unfairness, because in some scenarios, the sensitive attributes affect the decision along multiple pathways, and not all pathways are unfair. For example, in the loan application Zhang et al. (2017) shown in Figure 1, race (a sensitive attribute $R$) is only allowed to affect the loan application results through the income, since it is reasonable to reject a loan application due to the low income. Under this fairness rule, paths $R \rightarrow Y$ and $R \rightarrow Z \rightarrow Y$ are unfair paths and path $R \rightarrow Q \rightarrow Y$ is a fair path.

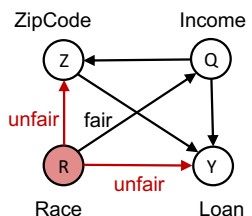

Figure 1: Loan Application Example.

To fulfill path-specific causal fairness, some existing works directly calculate the path-specific causal effect (PSE) Avin et al. (2005); Pearl (2001) along the unfair pathways, and minimize the effect simultaneously when maximizing prediction accuracy Wu et al. (2019b); Nabi & Shpitser (2018). Some other works correct the variables located on the unfair pathway by a latent inference-projection method Chiappa (2019). However, these existing works still face the following challenges: (1) Most of them require a pre-defined graph as the prior knowledge to calculate PSE. (2) The calculation of the path-specific effect is complex, requiring the sequential ignorability assumption Imai et al. (2010) to ensure the identification. (3) They all assume the sensitive attributes are root nodes in the causal graph. Namely, there are no other variables that affect the sensitive attributes. Few of them consider the case when the sensitive attributes are non-root nodes, which can be widely observed in real-world applications. For example, in the recommendation system, the item popularity is a sensitive attribute Ge et al. (2021); Zhu et al. (2018), while this variable is a non-root node as it is affected by the item's characteristics.

In light of the above challenges, we propose a **C**ausal **G**raph based **F**airness Framework, shortened as **CGF**. To tackle the first challenge about the lack of causal graph information, CGF integrates the causal graph structure learning and fair prediction, which reveals the causal relationships among the observed variables. To simplify the sophisticated PSE calculation, CGF imposes the fairness regularization at the graph level by restricting the existence of unfair edges in the learned causal structure. In this way, fair decisions are made based on the corrected observations reconstructed from the learned graph structure. Furthermore, the proposed CGF framework can straightforwardly generalize to the case where sensitive attributes are non-root nodes by introducing the latent variables to divide the fair and unfair effect flow. To the best of our knowledge, the proposed framework CGF is the first work considering such non-root node case.

## 2 RELATED WORK

Most of the existing works of path-specific causal fairness restrict the unfair pathways by reducing their path-specific effect. In Nabi & Shpitser (2018); Nabi et al. (2019), the prediction accuracy and the path-specific effect along with unfair causal pathways are jointly minimized. The work proposed in Zhang et al. (2017) designs a two-step algorithm, by first learning the graph structure and then minimizing the prediction error with PSE regularization. In Wu et al. (2019b), the authors adopt the response-function variable to bound the path-specific causal fairness. Instead of directly minimizing the path-specific effect, a latent inference-projection based method is proposed in Chiappa (2019) to correct the variables that are the descent nodes of sensitive attributes. In Helwegen et al. (2020), the CEVAE framework Louizos et al. (2017) is adopted to infer the causal mechanism based on the pre-defined causal graph, and then the auxiliary prediction model is constructed based on the selected causal relation along with the fairness requirement.

**Relations to Existing Work.** Most of the above existing works require the prior knowledge about causal graph to calculate the PSE or to correct sensitive variables' descent variables, which is hard to be satisfied in real-world applications. Compared with the work in Zhang et al. (2017) that has a separated time-consuming causal structure learning step, our work applies the fairness constraint on the continuous-optimization based graph structure learning, which can efficiently obtain the causal graph and simplify the PSE calculation. Furthermore, it is worth mentioning that all the above existing works assume that the sensitive attributes are root nodes. The proposed framework is the first work that generalizes to the case when sensitive attributes are non-root nodes under path-specific causal fairness. Additionally, our proposed framework is motivated by the work of utilizing the causal graph discovery to enhance the machine learning generalization ability Kyono et al. (2020). Compared with Kyono et al. (2020), the proposed CGF framework contains the cascade reconstruction step, which is the major difference. With the cascade reconstruction step, the unfairness contained in the original data can be corrected. Besides, CGF also has the fairness regularization in our proposed method, which reduces the unfair paths in the causal graph and meanwhile assures that the data correction follows the fair graph.

## 3 BACKGROUND

**Causal Graph.** A causal graph is a directed acyclic graph (DAG) reflecting the causal relationships between variables. Let $\mathcal{G}$ denote a causal graph, and $\mathcal{G} = \langle V, E \rangle$, where $V$ is the set of nodes representing all the variables, and $E$ is the set of edges with each edge $V_i \to V_j$ describing the causal relation between variable $V_i$ and $V_j$. The *parents nodes* of node $V_i$, denoted as $\Pi(V_i)$, and $V_j \in \Pi(V_i)$ if $V_j \to V_i$. A node is a root node if it has no parent nodes. A *path*, also named as causal pathway, is defined as a sequence of unique nodes with edges between each consecutive node. The *depth* of a node in the graph is the number of arrows in the longest path to the root nodes. In the rest of the paper, we use the term "node", "variable", and "attribute" interchangeably.

**Path-specific Causal Fairness.** Path-specific causal fairness ensures that the sensitive attributes are not allowed to affect the prediction along the unfair causal pathway. From the definition, path-specific causal fairness distinguishes the causal pathways that start from sensitive variables to predicted variables into fair paths and unfair paths, and the goal of fair prediction is to reduce the unfair paths.

**Relations to Other Fairness Definition.** Path-specific causal fairness is closely related to other definitions of fairness. It is equivalent to removing the direct and indirect discrimination Zhang et al. (2017). When all paths starting from the sensitive variables are unfair, achieving path-specific causal fairness is equal to demographic parity (i.e., removing disparate impact) Zafar et al. (2015).

**Structure Causal Model (SCM).** In Structure Causal Model(SCM), each node in $\mathcal{G}$ is associated with a causal mechanism representing the relation between the current node and its parent nodes. It is defined as: $\mathcal{F} = \{f_i : V_i = f_i(\Pi(V_i)) + \epsilon_i\}$, where $V_i \in V$ is the $i$-th node in the graph, $\Pi(V_i)$ is the set of parent nodes of $V_i$, and $\epsilon_i$ is the random noise.

**Definition 3.1.** *(Observed Graph). Observed graph is the causal graph of the observed data.*

**Definition 3.2.** *(Fair Graph). The causal graph satisfying the fairness criterion, and meanwhile, preserving the remaining structure of the observation graph, is the fair graph.*

**Definition 3.3.** *(Model Graph). Model graph is the causal graph that the decision model relies on.*

Figure 2 shows the observational graph, fair graph and model graph of the loan example in Section 1. Figure 2a is the observational graph, which is the causal graph of the observed data. In the graph, the fair path $R \to Q \to Y$ represents that it is acceptable, in terms of income, that some people with certain race have a lower loan approval rate because they tend to be underpaid. While, the paths $R \to Y$ and $R \to Z \to Y$ are unfair, indicating that it is disallowed that the race affects the loan approval directly or indirectly through ZipCode. Figure 2b is the fair graph, which describes the ideal causal relations. Compared with the observational graph, it eliminates the unfair paths. By removing the unfair paths, the fair graph reflects that the difference of loan application results across different race groups is explained by the different income levels among those groups. The rightmost sub-figure is the model graph, which is the graph that the model relies on to predict. As shown in Figure 2c, the model takes $R$, $Z$, and $Q$ as input, therefore, they all have directed arrows pointing to prediction $Y$.

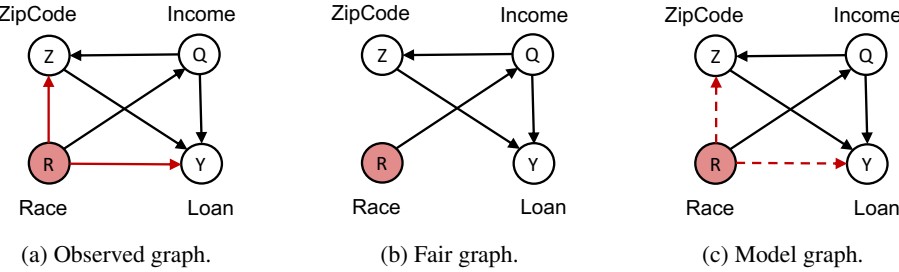

Figure 2: Causal Graphs of Loan Example.

From the above triple-graph perspective, under path-specific causal fairness, the model graph should be consistent with the fair graph, but it is not. Therefore, our objective is to exclude the unfair path (the red dashed arrow in Figure 2c) for the decision, and retain the remaining causal pathways.

## 4 METHODOLOGY

To satisfy the path-specific causal fairness, we propose a causal graph based fairness framework CGF, which imposes fairness at the graph structure level. The key of CGF framework is to make the causal graph and the data that the model relies on close to the ideal fair graph. In detail, the proposed framework contains three components: graph structure learning, fair regularization, and label prediction. The graph learning part aims to reveal the graph structure of the observation data, the fairness restriction targets at reducing the unfair paths, and the label prediction part outputs the fair predictions. These three components influence each other in that: the fairness restriction guides the graph learning by reducing the unfair edges, and in label prediction, the final prediction is made based on the values of its parent nodes which is detected by the graph learning component. Overall, the objective function is:

$$\mathcal{L} = L_{GL} + L_F + L_P, \tag{1}$$

where $L_{GL}$ is the graph learning loss, $L_F$ is the fairness restriction, $L_P$ is the label prediction loss. In the following sections, we will first present the detailed implementation when the sensitive attributes are root nodes, provide the theoretical analysis about the generalization error, and then generalize the developed method to the case where sensitive attributes are non-root nodes.

### 4.1 CGF FRAMEWORK

Under path-specific fairness criterion, when sensitive attributes are root nodes, such as age, gender or race, we need to reduce the unfair edges. In the following three subsections, we will present the implementations of the three components in Eqn. (1).

#### 4.1.1 GRAPH STRUCTURE LEARNING

The objective of graph structure learning is to find the optimal causal graph that fits the observed data best. Motivated by the continuous optimization of causal graph structure learning Zheng et al. (2018), the loss of graph structure learning is:

$$L_{GL} = \beta||D - \tilde{D}||_2^2 + \gamma_1 \left( tr(e^{W \odot W}) - (d_A + d_X + 1) \right)^2 + \gamma_2||W||_1, \tag{2}$$

where $W \in \mathbb{R}^{(d_A+d_X+1) \times (d_A+d_X+1)}$ is the adjacency matrix, and if its element $w_{i,j} \neq 0$, there exists an edge $V_j \rightarrow V_i$ with weight $w_{i,j}$ indicating the effect strength. $d_A$ is the dimension of sensitive attributes, and $d_X$ is the dimension of other features. $D$ is the observed data, and $\tilde{D}$ is the reconstructed data based on $W$ and $D$ according to the causal graph. $\odot$ is the element-wise matrix multiplication operator, $e^{W \odot W}$ denotes the matrix exponential of $W \odot W$, $tr(\cdot)$ is the matrix trace. $\beta$, $\gamma_1$, and $\gamma_2$ are the hyper-parameters.

The first term in Eqn. (2) measures the fitness of the causal graph by calculating the difference between the observed data and the data reconstructed from the graph. The second term is the directed acyclic graph (DAG) constraint, which ensures the learned graph does not contain cycle Zheng et al. (2018). The third term is the $\ell_1$ norm of the adjacency matrix which makes the learned graph to be sparse. The details of data reconstruction (i.e., $\tilde{D}$) and the DAG constraint are described as follows.

**Cascade Data Reconstruction.** In data reconstruction, each node is reconstructed based on its parents' reconstructed value. Eqn. (3) shows the reconstruction of a single node $V_i$:

$$\hat{V}_i = f_i(\hat{\Pi}(V_i)W[i_\Pi, i]), \tag{3}$$

where $f_i(\cdot)$ is the causal mechanism of node $V_i$, $\hat{\Pi}(V_i)$ is $V_i$'s parent nodes after reconstruction. $i_\Pi$ is the index set of $V_i$'s parent nodes. $W[i_\Pi, i]$ is the elements in the adjacency matrix $W$ whose row indices are in $i_\Pi$ and column indices are $i$. It is noticed that the reconstruction of a node is based on its parents' reconstructed values, instead of the observed values, because the observed values contain unfair effect if the parent nodes locate in the unfair paths. To satisfy this, the parent nodes should be reconstructed before the child nodes, thereby the data reconstruction follows a cascade reconstruction procedure with ascending order of depth.

We use a graph with five nodes, shown in Figure 3, to illustrate the cascade data reconstruction. In the figure, $A$ denotes the sensitive attribute, $X_2$, $X_3$, $X_4$ are regular features, and $Y$ is the class label. The red arrows denote the unfair edges. The reconstruction order decided by the depth is: $A$ & $X_4$, $X_2$, $X_3$, $Y$ and the reconstruction procedure is:

Root Nodes: $\hat{A} = A;\ \hat{X}_4 = X_4;$

Depth 1 node: $\hat{X}_2 = w_{1,2}A + w_{4,2}X_4 + b_2;$

Depth 2 node: $\hat{X}_3 = w_{1,3}A + w_{4,3}X_4 + w_{2,3}\hat{X}_2 + b_3;$

Depth 3 node: $\hat{Y} = w_{1,5}A + w_{4,5}X_4 + w_{3,5}\hat{X}_3 + b_Y,$

$$(4)$$

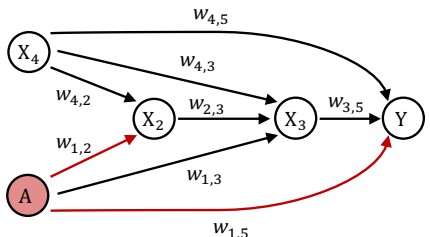

where $w_{i,j}$ is the element of the $i$-th row and the $j$-th column in adjacency matrix $W$, $b_i$ is the intercept term, and $\tilde{D} = [\hat{A}, \hat{X}_2, \hat{X}_3, \hat{X}_4, \hat{Y}]$. In this example, we adopt the linear function as the causal mechanism, and it can generalize to the more complex functions, such as neural network by changing $f_i(\cdot)$ in Eqn. (3).

Figure 3: Cascade Data Reconstruction Example.

**DAG Constraint.** The second term in Eqn. (2) is the directed acyclic graph (DAG) constraint, which ensures that there is no cycle in the learned graph Zheng et al. (2018), as a node cannot affect itself. The trace of adjacency matrix's exponential is adopted in the second term to measure the graph acyclic. The matrix exponential is given by the power series: $e^{W \odot W} = \sum\limits_{k=0}^{\infty} \frac{1}{k!}(W \odot W)^k$, where the $k$-th term denotes the adjacency after $k$ times walking on the graph, and $W \odot W$ makes the element of adjacency matrix non-negative. The trace of the $k$-th term ($k > 0$) should be zero if the graph is acyclic, because a node cannot go back to itself after $k$ times walking. Therefore $tr(e^{W \odot W}) = (d_A + d_X + 1)$ indicates that the graph is acyclic, where $d_A + d_X + 1$ is the trace of first term in the power series.

### 4.1.2 FAIRNESS REGULARIZATION

The goal of fairness regularization is to reduce the unfair edges, so that the sensitive attributes pass less effect through the unfair path. As mentioned previously, the element in the adjacency matrix not only represents the causal direction, but also indicates the effect strength along this edge. To reduce the unfair edges, the elements in the adjacency matrix associate with unfair edges should be close to zero. Therefore, we apply the following fairness regularization on the adjacency matrix:

$$L_F = \alpha ||W \odot M_F||_1, \tag{5}$$

where $\odot$ is the element-wise matrix multiplication, $W$ is the adjacency matrix, and $M_F$ is the fairness mask with the same dimension as $W$. The element of the $j$-th row, $i$-th column of $M_F$ is set as 1 if edge $V_i \rightarrow V_j$ is unfair. More details related to the construction of the fairness mask are in the appendix. $|| \cdot ||_1$ is the $\ell_1$ norm. The fairness regularization $L_F$ minimizes the total strength of effect on the unfair edges, which reduces the effect flow along the unfair paths. By regularizing on the adjacency matrix, fairness regularization is able to eliminate the unfair path $A \rightarrow Y$ and $A \rightarrow X_2$ in Figure 3. Therefore, with the fairness regularization, the reconstructed data $\tilde{D}$ is a correction of the original data $D$ with unfair effect reduced.

### 4.1.3 LABEL PREDICTION

Since the unfair effect has been reduced in the reconstruction data, the prediction based on the obtained reconstruction is fair. The label $Y$ is predicted as: $\hat{Y} = \tilde{D}W_Y$, where $W_Y$ is the last column of $W$, which indicates the existence of the edges and their effect strength starting from other nodes to $Y$ and $\tilde{D}$ is the reconstructed data. Accordingly, the prediction loss is: $L_P = ||Y - \hat{Y}||_2^2$.

### 4.1.4 GENERALIZATION TO NONLINEAR CAUSAL MECHANISM.

In Eqn. (4) and label prediction, the adopted linear causal mechanism can generalize to more advanced functions by modify the cascade data reconstruction and the adjacency matrix. Assume we choose neural network (NN) as the causal mechanism, the reconstruction of node $V_i$ is: $\hat{V}_i = f_i^{NN}(DW_i^{NN})$, where $f_i^{NN}(\cdot)$ represents the neural network, $W_i^{NN} \in \mathbb{R}^{(d_A+d_X) \times d_{NN}}$ is the parameter of the first linear layer in $f_i^{(\cdot)}$, $d_{NN}$ is the dimension of the first hidden layer. Namely, to use the NN mechanism, replace the linear model in Eqn. (4) with the NN whose first layer is the linear layer and all those

NNs share the common hidden layers, as suggested in Kyono et al. (2020). The adjacency matrix is constructed based on the parameter in the first linear layer. Specifically, each element in the adjacency matrix $W$ is calculated as: $w_{i,j} = ||W_i^{NN}[j,:]||_2^2$, where $W_i^{NN}[j,:]$ is the $j$-th row in $W_i^{NN}$.

## 4.2 THEORETICAL ANALYSIS

Here we provide the theoretical analysis about the generalization error of the proposed framework.

**Theorem 4.1.** *Suppose the the data $D$ follow the Gaussian distribution, the generalization error of the proposed fair classifier $h_F$ on the observed dataset, which is denoted as $\epsilon_{h_F}^{\mathcal{D}^{ob}}$, satisfies the following inequality with probability $1 - \delta$, $\forall \delta > 0$:*

$$\epsilon_{h_F}^{\mathcal{D}^{ob}} \leq \left( ||D - \tilde{D}||_{fro}^2 + \tau \sqrt{\frac{2 \log \frac{2}{\delta}}{m}} + \frac{C_1}{\sqrt{m\delta}} + \frac{C_2}{\sqrt[4]{m\delta}} + C_3 \right)^{\frac{1}{2}}$$
$$+ 4 \hat{\epsilon}_{h_F}^{\mathcal{D}^F} + \frac{1}{m} \left[ \mathcal{R}_{dag} + C_4 (\mathcal{R}_{l_1} + \mathcal{R}_F) + \log(\tfrac{8}{\delta}) \right] + C_5,$$

(6)

*where $\mathcal{D}^{ob}$ is the distribution of the observed data and $D$ is its observed sample. $\tilde{D}$ is the reconstructed data by cascade reconstruction, and its distrubution is denoted as $\mathcal{D}^F$. $\epsilon_h^{\mathcal{D}} = \int_{\mathcal{D}} \ell_h(a, x) p^{\mathcal{D}}(a, x) da dx$, is the expected error on the underlying space $\mathcal{D}$, and $\ell_h(a, x) = \int_Y L(Y, h(a, x)) p(Y|a, x) dY$ is the expected error on a single point $(a, x)$. $\hat{\epsilon}_{h_F}^{\mathcal{D}^F}$ is the empirical error of classifier $h_F$ on $\mathcal{D}^F$. $\mathcal{R}_{dag}$ and $\mathcal{R}_{l_1}$ are the value of DAG constrain, $\ell_1$ regularization, which are the last two terms in Eqn. (2). $\mathcal{R}_F$ is the value of fairness regularization defined in Eqn. (5). $\tau$, $C_1 \sim C_5$ are constants.*

Theorem 4.1 give an upper bound of the generalization error of the fair classifier on the observed dataset. The upper bound shows that the generalization error is related to the qualities of the reconstruction and the classifier trained on the fair dataset, which is exactly the two terms in our objective function. The reconstruction part in Thm 4.1 also represents the fairness level, since the fairer the data is, the fairer the dataset is, the smaller the reconstruction error

## 4.3 GENERALIZATION TO NON-ROOT NODE CASE

Most of the existing works consider sensitive attributes such as age, gender, and race that can only be the root nodes in the causal graph. However, in some real-world applications, sensitive attributes are affected by other variables. For example, in the recommendation system, the item popularity should not affect whether this item to be recommended Ge et al. (2021); Zhu et al. (2018), for the purpose of recommendation diversity. Figure 4 shows causal graph of the recommendation example where $U$ and $I$ represent user and item respectively, $P$ denotes the item popularity and $Y$ denotes whether the user click the item. In this example, the item popularity $P$ is the sensitive attribute, and it is the non-root node as it is affected by the item's characteristics.

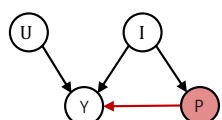

Figure 4: Sensitive Attributes as Non-root Nodes.

**Challenge.** When the sensitive attributes $A$ are non-root nodes, their parent nodes $\Pi(A)$ also contain the information about those sensitive nodes. If the parent nodes have other causal pathways to the label node $Y$ not passing the sensitive node, the information related to the sensitive attributes can still reach the label node through those paths. Therefore, the proposed framework in Section 4.1 requires slight modification to handle this case.

**Effect Diversion.** To tackle the above challenge, we add two latent nodes between the parent nodes and the sensitive nodes to divert the effect flow, so that the fairness regularization proposed in the previous section can be applied to the unfair flow. One latent node, denoted as $Z_s$, controls the effect from sensitive attributes' parent nodes to label node passing through sensitive attributes. The other latent node $Z_y$ controls the effect from parent node to label node not passing through sensitive attributes. In other words, the paths from parent nodes $\Pi(A)$ to label node $Y$ are divided into two categories, one only contains latent node $Z_y$, and the other only contains $Z_s$. By disentangling $Z_y$ and $Z_s$, the fairness regularization now can be directly applied to the paths containing $Z_s$, as no other paths face information leakage risk. When there are multiple non-root sensitive attributes, the above procedure is applied to each one of them.

Figure 5 shows the causal graph with effect diversion in the recommendation example, where the latent node $Z_u$ is user embedding, $Z_s$ is the item popularity related embedding and $Z_y$ is the clicking variable $Y$ related embedding. The parent node $I$ affects item popularity only through $Z_s$, and other paths from $I$ to $Y$ all pass through $Z_y$. The red arrows indicate the unfair paths where the fairness regularization is applied on.

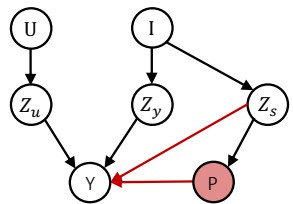

Figure 5: Causal Graph with Effect Diversion.

**Objective Function.** In Eqn. (2), the reconstruction part requires the variables' observed values, while in this case latent nodes lack that. To address this issue, we notice that to fit the graph with effect diversion, $Z_y$ and $Z_s$ should have less overlapped information. To fulfill this, the orthogonal regularizations on $Z_y$ and $Z_s$ are adopted in the graph structure learning part to ensure the separation of the effect flow. Overall, the objective function is:

$$\mathcal{L} = ||Y - \hat{Y}||_2^2 + \beta||D - \tilde{D}||_2^2 + \beta_z \sum_{i=1}^{\mathcal{I}} cos(Z_s^{(i)}, Z_y^{(i)})$$
$$+ \gamma_1 \left( tr(e^{W \odot W}) - (d_A + d_X + 1) \right)^2 + \gamma_2 ||W||_1 + \alpha ||W \odot M_F||_1, \tag{7}$$

where $\mathcal{I}$ is the number of total items. $cos(\cdot, \cdot)$ denotes cosine similarity, which ensures the orthogonality between $Z_s^{(i)}$ and $Z_y^{(i)}$, and other types of correlation measure such as HSIC Gretton et al. (2007) can be adopted. Compared with Eqn. (1), the graph structure learning part is modified to satisfy the effect diversion design. In the recommendation system example, $||Y - \hat{Y}||_2^2$ is the clicking prediction error, and $||D - \tilde{D}||_2^2$ is the error of predicting the item popularity.

## 5 EXPERIMENT

We experiment on both synthetic and real-world datasets to confirm: (1) The proposed CGF framework works on both cases where sensitive attributes are root or non-root nodes. (2) Our proposed framework provides a better trade-off between utility and fairness. More details regarding CGF implementations, experiment settings and additional experiment on synthetic dataset are listed in the Appendix.

### 5.1 EXPERIMENT ON ADULT DATASET

**Dataset.** Adult dataset[1], is a commonly adopted dataset for fairness evaluation. In this dataset, there are total $48842$ individuals and each has $14$ attributes regarding their demographic information, jobs, and level of education. The class label is binary indicating whether an individual's income is above or below $50k$. The objective is to predict the income class given an individual's attributes.

**Unfair Paths.** As suggest in Nabi & Shpitser (2018), the direct path "Gender" $\rightarrow$ "Income", and the paths containing edge "Gender" $\rightarrow$ "Married" are all unfair. Namely, the gender should not direct affect the income and meanwhile it is not allowed to affect income through marital status.

**Baselines.** The logistic regression model (LR) and neural network (NN) constructed from raw data is adopted as the baseline. We also adopt the Fair Inference (FIO) Nabi & Shpitser (2018) and PSE-DR Zhang et al. (2017) as the baselines. Our proposed models are denoted as LR-CGF and NN-CGF, which take the linear logistic regression model and neural network as the causal mechanism function, respectively.

**Evaluation Metrics.** Due to label imbalance, Area Under the ROC Curve (AUC) is adopted as the utility metric. The higher the AUC value, the better the utility. Following Nabi & Shpitser (2018), we adopt the path-specific causal effect (PSE) Pearl (2001); Shpitser (2013) to measure the fairness. The PSE value may have a negative value indicating the negative effect. The closer the PSE value is to $0$, the fairer the model is.

### 5.1.1 RESULT ANALYSIS

Table 1 summarizes the results of different methods on Adult dataset with 5-fold cross validation. It is observed that compared with baseline methods, our proposed methods are fairer and meanwhile

---

[1]https://archive.ics.uci.edu/ml/datasets/adult

have better utility. We also notice that the method with nonlinear causal mechanism performs best in terms of both utility and fairness. The reason is that compared with linear causal mechanism, the neural network can reconstruct the data better while ensuring fairness. This observation also confirms Theorem 4.1 that the better the reconstruction is, the better the performance is.

We further experimentally explore the relationship between the reconstruction and fairness regularization. We fix one part's hyperparameter and tune the other one. Figure 6 reports the results of NN-CGF. From Figure 6a and 6b, it is observed that, with the increasing strength of reconstruction, it improves the accuracy but reduces the fairness. The fairness regularization has an opposite effect with reconstruction part. As shown in Figure 6c and 6d, the fairness regularization

| Method | AUC ($\Uparrow$) | PSE ($\Rightarrow 0$) |
|--------|------------------|----------------------|
| LR | $0.712 \pm 0.005$ | $3.508 \pm 0.005$ |
| NN | $\mathbf{0.721 \pm 0.012}$ | $2.068 \pm 0.223$ |
| FIO | $0.505 \pm 0.007$ | $1.048 \pm 0.003$ |
| PSE-DR | $0.686 \pm 0.018$ | $0.450 \pm 0.151$ |
| LR-CGF | $0.507 \pm 0.099$ | $0.925 \pm 0.073$ |
| NN-CGF | $\mathbf{0.689 \pm 0.012}$ | $\mathbf{-0.198 \pm 0.109}$ |

Table 1: Results on Adult Dataset. $\Uparrow$: the higher the better, and $\Rightarrow 0$: the closer to 0, the better.

improves the model fairness but reduces utility. Overall, the reconstruction part and the fairness regularization, together, control the trade-off between model utility and fairness.

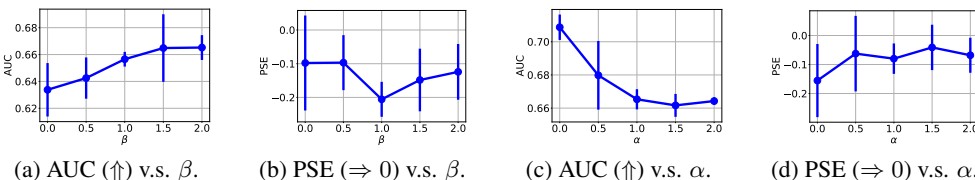

(a) AUC ($\Uparrow$) v.s. $\beta$.    (b) PSE ($\Rightarrow 0$) v.s. $\beta$.    (c) AUC ($\Uparrow$) v.s. $\alpha$.    (d) PSE ($\Rightarrow 0$) v.s. $\alpha$.

Figure 6: Effects of Reconstruction and Fairness Regularization.

## 5.2 EXPERIMENT ON RECOMMENDATION DATASET

**Dataset.** The MovieLens dataset Harper & Konstan (2015) is adopted to validate the performance of CGF. Following the settings in Ge et al. (2021), the sensitive attribute *item popularity* is added to each item, and for each item, the value of item popularity is 1 if its total exposure is top $20\%$, otherwise 0. The item popularity is a non-root node since it is affected by item characteristics. For each user, we sort their interactions according to the timestamp, and the last interaction is put into the test set, and others are in the training set. The validation set is the last interaction of each user in the training set.

**Baselines.** We adopt Matrix Factorization (MF) Koren et al. (2009), Generalized Matrix Factorization (GMF) and Multiple Layer Perceptron (MLP) He et al. (2017) as our baselines. GMF and MLP are also the base models of our proposed framework, and we name our methods as GMF-CGF and MLP-CGF accordingly.

**Evaluation Metrics.** In terms of utility/accuracy measure, Top-k ranking metrics hit rate (HR) and normalized discounted cumulative gain (NDCG) are adopted to measure the recommendation performance. Following Ge et al. (2021), the Gini Index and Popularity Rate (PR) are also adopted to measure the fairness. The details of Gini Index and Popularity Rate (PR) are in the Appendix. For HR and NDCG, the higher the value is, the better the performance is. For Gini Index and PR, the lower the value is, the fairer the model is.

### 5.2.1 RESULTS ANALYSIS

Table 2 shows the results of baselines and our proposed methods with Top 10 rankings metrics. We also list the results of our methods' variants. $*$-CGF w/o rec denotes the CGF without the orthogonal regularization part, i.e., $\beta_z = 0$ in Eqn. (7). $*$-CGF w/o fairness and denotes CGF without the fairness regularization part ($\alpha = 0$). $*$-CGF w/o ord & fairness is CGF without both of these two parts.

In terms of fairness, our proposed methods recommend more diverse items, and meanwhile, have the comparable recommendation accuracy to the baselines. This observation verifies that our proposed method makes the base model to be fair without scarifying too much utility. It is worth to mention that the results measured by GINI and PR are also the indirect indicator of how good the disentanglement of the effect from sensitive attributes' parent nodes to label node. The better it disentangles, the fairer the model. Furthermore, the ablation results shown in the table 2 indicate that the orthogonal regularization and the fairness regularization both contribute to the model fairness.

| Method | HR (⇑) | NDCG (⇑) | GINI (⇓) | PR (⇓) |
|---|---|---|---|---|
| MF | **0.197** | **0.103** | 0.878 | 0.861 |
| GMF | 0.195 | 0.098 | 0.875 | 0.857 |
| MLP | 0.149 | 0.077 | 0.919 | 0.925 |
| GMF-CGF (ours) | 0.166 | 0.084 | **0.837** | **0.773** |
| GMF-CGF w/o ord | 0.174 | 0.089 | 0.853 | 0.820 |
| GMF-CGF w/o fairness | 0.166 | 0.089 | 0.857 | 0.808 |
| GMF-CGF w/o ord & fairness | 0.195 | 0.102 | 0.881 | 0.860 |
| MLP-CGF (ours) | 0.116 | 0.059 | 0.882 | 0.844 |
| MLP-CGF w/o ord | 0.112 | 0.054 | 0.902 | 0.891 |
| MLP-CGF w/o fairness | 0.141 | 0.066 | 0.923 | 0.932 |
| MLP-CGF w/o ord & fairness | 0.139 | 0.066 | 0.903 | 0.863 |

Table 2: Results on the Movielens Dataset. ⇓ indicates the lower, the better.

To further analyze the effect of orthogonal regularization and fairness regularization, in Figure 7, we plot the four metrics with respect to different regularization strengths by tuning one hyper-parameter and fixing the others. From this figure, we can observe that the stronger the regularization strength is, the fairer the model is, and the more utility is sacrificed. Furthermore, The utility and fairness trade-off can be controlled by tuning the values of two regularizations' hyper-parameters. We also notice that MLP-CGF performs slightly different in terms of HR and NDCG: The stronger the orthogonal regularization and the fairness regularization, the better the performance. The reason is that compared with GMF-CGF, MLP-CGF has more learnable parameters in the neural network, and adding those regularizations would prevent the over-fitting.

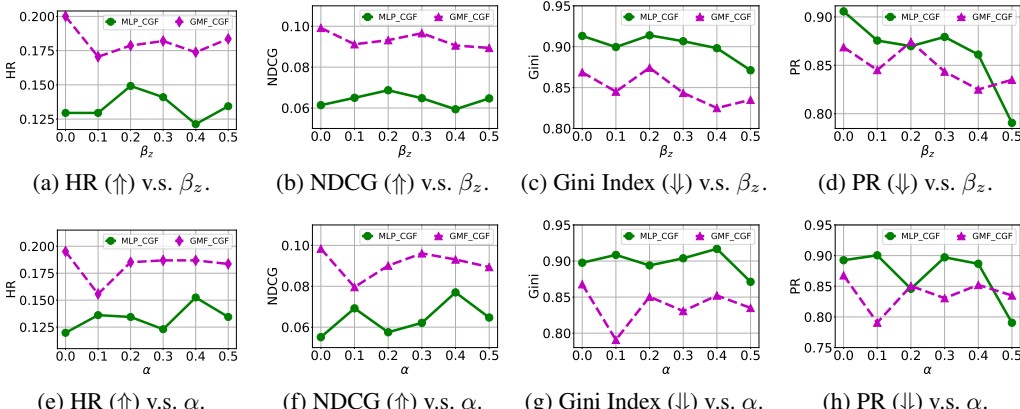

(a) HR (⇑) v.s. $\beta_z$.    (b) NDCG (⇑) v.s. $\beta_z$.    (c) Gini Index (⇓) v.s. $\beta_z$.    (d) PR (⇓) v.s. $\beta_z$.

(e) HR (⇑) v.s. $\alpha$.    (f) NDCG (⇑) v.s. $\alpha$.    (g) Gini Index (⇓) v.s. $\alpha$.    (h) PR (⇓) v.s. $\alpha$.

Figure 7: Effects of the Orthogonal Regularization and Fairness Regularization.

## 6 CONCLUSIONS AND FUTURE WORK

In this work, we propose a novel causal graph based fair prediction framework under path-specific causal fairness. The core of the proposed framework is to ensure that the graph adopted by the prediction model should be close to the fair graph. To fulfill this, we integrate the graph structure learning and the fairness regularization in an interactive way. The learned graph structure reveals the causal graph of the original observations with unfair edges eliminated, and the data reconstructed from the learned graph is close to the original observations with unfair effect corrected. Based on the corrected causal graph and its associated data, the prediction model achieves the path-specific causal fairness. Experimental results on the real-world dataset confirm that the proposed framework ensures fair predictions and meanwhile retains the comparable utility. We also generalize the proposed framework to the case of sensitive attributes being non-root nodes by effect redividing, which is further validated by experiments on a real-world recommendation dataset.

In this paper, we assume that there are no latent confounders in the dataset. When this assumption is not satisfied, the causal graph may not be identified from the observation data. Recently, some causal discovery works that target to recover the causal graph in the presence of latent confounders Xie et al. (2020); Cai et al. (2019) have been developed. Relaxing the no latent confounders assumption and generalizing our work to the latent confounders case will be the future work.

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
