# OpenReview forum: "Path-specific Causal Fair Prediction via Auxiliary Graph Structure Learning"
_ICLR.cc/2022/Conference — ICLR 2022 Submitted_

### Official Review · Reviewer_nUyn · 2021-11-01

**Correctness:** 3
**Technical Novelty And Significance:** 2
**Empirical Novelty And Significance:** 2
**Recommendation:** 3
**Confidence:** 5

**Main Review:**

Strength:

1. This paper targets challenging problems in causal fair machine learning: the causal graph is required, and the computation of PSE is complex.
2. The idea of combining causal structure learning and causal fairness regularization is novel.
3. The experimental results show the effectiveness of the proposed method, compared with the baseline approaches.

Weakness:

1. Some summaries of the existing research are inaccurate. The ignorability assumption or identification requirement is not necessary. [1] provides a bounded method for unidentification cases. The sensitive attributes are not required as the root nodes. Path-specific effects are generic as defined by Avin et al. 2005. Zhang and Bareinboim 2018b define spurious discrimination which covers the effect from the sensitive attribute to the decision via a confounder.

2. The definition of the Structure causal model: the function is not necessarily an additive-noise style.

3. Eq (5): there is a gap between fair edges and fair paths, further small path-specific effects. Intuitively, the small elements in an adjacent matrix lead to small effects but it is not necessary. In addition, the mask construction described in the Appendix is not clear. The path-specific effect is defined via paths. But the mask matrix is edge-based. There is a gap between the path and the edge-based definitions. For example, the mask considers paths from R to Y are fair iff the first edge is R->Q along the paths. It considers R->Q->Y or R->Q->O->Y as fair but R->O->R->Y as unfair, which is not convincing.

4. Ch. 4.3: the spurious effect from P<-I->Y is not unfair based on the definition of path-specific fairness as changing P doesn't affect Y. In addition, if there is no causal graph, it is unclear how to distinguish the root-node cases and non-root-node cases. In the experiments, the root/non-root cases are separated based on domain knowledge but causal discovery is able to find confounders beyond the domain knowledge.

5. It is not clear how the proposed method addresses the challenge of unidentification.

6. Fig 6 shows alpha is tuned based on the PSE value. It is expected that the path-specific effect is not used as CGF is proposed to address the challenge of complex computation of PSE and simply the PSE calculation.

[1]L. Zhang, Y. Wu, and X. Wu, “Causal modeling-based discrimination discovery and removal: Criteria, bounds, and algorithms,” IEEE Trans. Knowl. Data Eng, 2019, doi: 10.1109/TKDE.2018.2872988.



**Summary Of The Paper:**

This paper proposes a framework to integrate graph structure learning and path-specific fairness constraints into prediction tasks. The loss function consists of three parts: the graph structure learning loss through cascade data reconstruction, the causal fairness regularization through an adjacency matrix, and the label prediction.

**Summary Of The Review:**

The idea of combining structure learning and path-specific effect regularization is very interesting. But the gaps/incorrectness concerns me most.

Based on my understanding, the three challenges are not well addressed:

1. The requirement on causal graphs: The proposed framework does not require the causal graph but it cannot correctly define the path-specific effect or readily recognize the root/non-root cases. The proposed approach faces the challenges of unidentidication.

2. The complex calculation of PSE: PSE is calculated for parameter tuning.

3. The root assumption is not necessary for PSE. In addition, the proposed method requires domain knowledge to recognize the root/non-root case.

---

### Official Review · Reviewer_V6if · 2021-11-02

**Correctness:** 3
**Technical Novelty And Significance:** 2
**Empirical Novelty And Significance:** 2
**Recommendation:** 3
**Confidence:** 4

**Main Review:**

I have several concerns in terms of its technical contributions, clarity and experiments. Please see them as follows.

Concerns
- The technical contributions seem a bit limited. This work is largely relied on the existing work by Kyono et al., including a majority part of the loss function as well as a majority part of the generalization error analysis. The major contribution lies in the cascade data reconstruction.

- Since I am not expert in causality, please clarify if this concern makes sense. If we have a large causal graph, let's say n nodes in the causal graph, then we need n different causal mechanisms. I am afraid it won't scale up, especially when we are using a NN as a causal mechanism.

- In section 4.1.4, there is a typo regarding the first linear layer. Is there any intuition of setting w_ij as the squared l2 norm? And why are we only using the parameter in the first linear layer?

- I have several questions in terms of the experiment: (1) Why cannot we have a method for MF (like MF-CGF)? (2) Since you used GMF and MLP in He et al., why is its final model NeuMF not considered as a baseline? (3) I think it is a bit overclaim about "This observation verifies that our proposed
method makes the base model to be fair without scarifying too much utility." because MLP-CGF has a much worse utility compared with MLP.

- In appendix about adjacency matrix construction, it is unclear why the entry in adjacency matrix is calculated in this way. Please explain the motivation or intuition here.

**Summary Of The Paper:**

This paper aims to improve path-specific causal fairness by removing unfair causal pathways. Basically, the framework leverages the existing work by Kyono et al. and adds additional regularization terms. A big advantage of the proposed method is that it can handle the case where sensitive attribute is not root node. Based on experimental results, it seems CGF achieves a good trade-off in fairness and quality.

**Summary Of The Review:**

I think this work largely relies on an existing work, which limits its technical contribution. The paper clarity can be further improved by giving more intuitions or explanations to some tricks (or definitions maybe). As such, I think this paper could be much further improved.

---

### Official Review · Reviewer_ZYeg · 2021-11-09

**Correctness:** 3
**Technical Novelty And Significance:** 2
**Empirical Novelty And Significance:** 2
**Recommendation:** 3
**Confidence:** 3

**Main Review:**

This is an interesting paper which tackles an important problem. I agree that the approach is interesting. But I feel the technical novelty is limited. I noticed that the authors used L_{GL} from Zhang et al. 2018 in Eq. (2) directly and added fairness aware loss L_F and label prediction loss L_P.  These losses are nothing new (I understand the fairness aware loss is blocking some path, but there is no technical novelty in that aspect).

I found the generalisation error analysis is a bit decorative. There is no comment of a wide variety of parameters like $\mathcal{R}_{\cdot}$. How do they look like? What are the typical values of these parameters? While for examples in euclidean space, these generalisation bounds are easy to understand, for learning with graphs, they are difficult and the authors should make a great effort to elaborate on this.

I have several reservations about the loss function L_{GL} *in the current context* :  First, why do you need to enforce sparsity constraint? Why is it being assumed that the underlying graph is sparse? Such constraints can block some necessary fair paths, right? Second, sparsity constraint in terms of L_1 regularizer can work well for large graphs.  I believe the number of the nodes in the graph is very small-- right? To my understanding, $|V|=14 $ for Audit dataset. Hence, I am not sure why do we need such constraint.
In contrast, what I would prefer is to have a dissimilarity regularizer between causal graph and the reconstructed graph in terms of Kernel measures for example. In fact, I am not sure how the euclidean distance $||D-\bar{D}||$ will enforce causality in reconstructed graph--- this is the observed data from the graph but not the graph itself and therefore, I am not sure it will be free from confounding effect.

Finally, I do think that the experiments should be improved. I did not found any ablation study on different loss components. Maybe I have overlooked?

The statement "Due to label imbalance, Area Under the ROC Curve (AUC) is adopted as the utility metric." It is misleading. For label imbalance AUC is rather high and over optimistic. For example, consider a ranked list of 1000 items with 100 positive and 900 negative items. Now if the first positive item appears in 100th place, AUC can be as high 0.88 which is extremely high for an algorithm which assigns the first positive item in 100th position. AUC often helps in training with the pairwise surrogate ranking loss. But for evaluation with imbalance data, average precision on positive and negative items are preferred.

Typo: Page 5, second last line: "linear layer in f_i..":  Please correct the typo on f_i


**Summary Of The Paper:**

This paper introduces fair prediction framework which combines graph structure learning into fair prediction to ensure  that unfair pathways are discouraged in the causal graph. It  shows generalization bound and proves the efficacy of the underlying method using several experiments.

**Summary Of The Review:**

Interesting direction, but lack of novelty. Still lot of work is needed to get a good score.

---

### Decision · Program_Chairs · 2022-01-20

**Decision:**

Reject

**Comment:**

In this paper, the authors aim to work within the path-specific framework to implement fair predictions by learning a causal graph in such a way that some path-specific effect is removed.

Generally, the paper was not received very well by reviewers, with the primary concern being lack of novelty in particular in comparison with (Kyono et al.)

One additional comment I wanted to make is this: any prediction task that is a part of a pipeline that contains a graphical model selection step is properly a "post-selection inference problem."  Such problems are very challenging because:

(a) Learning a graph from data is known to lack consistency at any rate (meaning that the algorithm is only pointwise consistent, but not uniformly consistent).  This issue propagates to "downstream" tasks in the pipeline, including prediction problems.  Probably, the way this issue would manifest in this work is unless sample sizes were very large, there would be no particular reason to assume the correct causal path is removed.

(b) Even if uniformly consistent modifications of structure learning algorithms were used, the uncertainty in learning the graph with error must be propagated to all subsequent steps in the pipeline.  Doing so appropriately is very challenging.

When revising the paper, in addition to taking reviewer comments into account, please consider how your method deals with post-selection inference issues -- I think this is a very interesting but challenging question that is likely to come in peer review.